# Spark Plasma Sintering of Electric Discharge Machinable 1.5Yb-1.5Sm-TZP-WC Composites

Ella Walter, Maximilian Rapp and Frank Kern *

Institute for Manufacturing Technologies of Ceramic Components and Composites,
Faculty 7—Engineering Design, Production Engineering and Automotive Engineering, University of Stuttgart,
Allmandring 7b, 70569 Stuttgart, Germany; ella.illenseer@gmail.com (E.W.);
maximilian.rapp@ifw.uni-stuttgart.de (M.R.)
* Correspondence: frank.kern@ifkb.uni-stuttgart.de

**Abstract:** Electrically conductive zirconia tungsten carbide composites are attractive materials for manufacturing precision components by electrical discharge machining due to their high strength, toughness and electrical conductivity. In this study, nanocomposite ceramics with a ytterbia samaria co-stabilized zirconia 1.5Yb-1.5Sm-TZP matrix and 24–32 vol.% tungsten carbide dispersion were manufactured by spark plasma sintering (SPS) at 1400 °C for 15 min at 60 MPa pressure. The materials exhibited high strengths of 1300–1600 MPa, a moderate fracture resistance of 6 MPa$\sqrt{m}$ and an ultrafine microstructure with grain sizes in the 150 nm range. Scanning electron microscopy and RAMAN spectroscopy revealed the in situ formation of carbon during the SPS process and carbon formation scales with tungsten carbide content, and this apparently impedes bending strength.

**Keywords:** zirconia; tungsten carbide; spark plasma sintering; electrical discharge machining; microstructure; mechanical properties; phase composition





## 1. Introduction

Electrically conductive and, thereby, electric discharge machinable ceramics for mechanical engineering application can be manufactured by consolidating mixtures of a structural ceramic matrix such as alumina, zirconia or silicon carbide or nitride and a percolating electrically conductive dispersion such as a transition metal carbide, nitride or boride, carbon nanotubes or graphene [1–7]. Material systems based on a matrix of zirconia stabilized in the tetragonal modification (TZP = tetragonal zirconia polycrystal) provide a combination of high strength and toughness due to the transformation toughening effect [8]. Fine tungsten carbide dispersions of 24–40 vol.% are known to provide sufficient electrical conductivity in the range of >20 kS/m to enable electric discharge machining (EDM) [9,10]. EDM is today widely applied to machine metals or cemented carbides for mechanical engineering applications, especially for applications in tools, dies or other customized mechanical engineering components of high geometrical complexity [11]. Ceramics are also interesting candidates for such applications due to their high hardness and chemical stability. As EDM is a contactless and force-free nonconventional machining technology capable to machine materials irrespective of high hardness and abrasion resistance, it is, therefore, a very attractive technology for ceramic materials [7]. As most ceramic compositions designed for EDM are, due to the pinning effect of the dispersion, not sufficiently sinterable, pressure assisted sintering processes such as hot isostatic pressing (HIP), hot axial pressing (HP), gas pressure sintering (GPS) or spark plasma sintering (SPS) are frequently applied to obtain dense and defect-free semifabricates [12].

In SPS, the powder mixture is enclosed in a graphite pressing die (as for HP). The process combines an axial compression process with simultaneous heating by pulsed electric current heating via the conductive upper and lower punch of the axial pressing setup. While the actual effects taking place during SPS are still under controversial discussion, it can be

summarized that a significant acceleration of diffusion processes takes place promoted by the pulsed electric current, which enables densification of samples at lower temperatures and at shorter dwells [13,14]. Tokita recently summarized the new developments in SPS technology, which not only enables consolidation of metals for e.g., sputter targets, but also functionally graded materials and binderless carbides [15–18]. SPS, when properly carried out, leads to dense materials and fine grain microstructure [19,20]. Due to the shortening of the process time (not only the sintering process as such but also the heating and cooling cycle) compared to HP, SPS may result in substantial savings compared to the HP process, which is currently the state-of-the-art for many EDM-ceramics. In some previous studies, the changes in the material properties of ED-machinable ZTA materials made by SPS and HP were analyzed. Materials do not only differ in grain size and mechanical properties, it was also found that the processing window for good quality material was significantly narrower for SPS than for than for HP. Moreover, the percolating conductive phase is preferentially heated during SPS, and this may lead to the formation of rigid scaffolds of the conductive phase, which may impede further densification [21,22].

Various studies on zirconia graphene composites have been published in the past years. However, the desired strengthening and toughening effect of the graphene dispersion to the TZP matrix has not always been achieved [23]. Gallardo reported on the importance of processing route, which has a strong influence on graphene dispersion [24]. The anisotropic orientation of graphene depending on shaping technology has also been reported. In HP and SPS, which are axial pressing processes, graphene orients preferentially orthogonally to the pressing direction, which results in composites with anisotropic properties [25]. The type of graphene raw materials is also relevant. Few layer graphene (FLG) based materials seem to be promising yet very expensive raw materials for TZP-Graphene composites, as these materials may bend along grain boundaries [26]. Obradovic reported on a progressive softening of TZP/Graphene materials based on increasing amounts of large rigid multilayer graphene nanoplatelets, which results in weak graphene/zirconia interfaces, thereby decreasing strength and toughness [23]. The in situ formation of graphene has been reported by Miranzo and Ünsal in the case of SPS sintered silicon carbide and boron carbide/titanium diboride ceramics [27,28]. In situ formation should lead to a very isotropic homogeneous distribution of graphene in the composite, thereby avoiding texture formation observed in pre-alloyed composites. In the case of ED-machinable zirconia based materials, the stabilizer recipe of the zirconia matrix has a strong influence on the transformability of the tetragonal phase, the achievable transformation toughness and thereby mechanical properties. Hence, it can be used to tailor the mechanical properties of TZP-WC composites. The critical parameters are the ionic radii of the stabilizer cations, their concentration and their distribution. 1.5Yb-1.5Sm costabilized-TZP is therefore a compromise between the state-of-the-art 3Y-TZP, which leads to composites with high strength but limited toughness, and 1.5Y-1.5Nd-TZP-based materials, which offer ultimate toughness but limited strength [29].

1.5Yb-1.5Sm-TZP-WC composites based on identical powder mixtures were previously manufactured by hot pressing and characterized with respect to ED-machinability mechanical and electrical properties, ED-machinability and microstructure [30]. In the present study, the exact same recipes were densified by SPS in order to check if materials with identical properties and EDM characteristics are obtained.

## 2. Materials and Methods

The recipes and the compounding procedure of the powder mixtures are explained in detail elsewhere. The materials of this study are prepared from the remaining stock of pressing feedstock of the HP study [30]. SPS sintering was carried out in a graphite die of 40 mm diameter. All areas in contact with the plate were protected by graphite paper to avoid sticking. Three samples of each recipe (24, 28 and 32 vol.% WC) of 4.2 mm thickness (sintered) were consolidated at 1400 °C at 60 MPa pressure and 15 min dwell. After the removal of graphite paper, the samples were preground to remove circumferential

grit and then automatically lapped with 15 μm diamond suspension on both sides. For a better understanding of how the samples were prepared and which measurements were made, an overview of the sampling scheme is given in Figure 1. One side of the disks was polished with 15 μm, 6 μ, 3 μm and 1 μm diamond suspension to a mirror-like surface (Struers Rotopol, Ballerup, Denmark). One disk of each recipe was kept for EDM tests. Two disks were cut into bending bars of 2 mm thickness. One side of the bars was polished to a mirror-like surface (as described before), and the back side was lapped with 15 μm suspension. The edges were carefully beveled to remove cutting defects. Density, ρ, was measured by the buoyancy method in air and water. Mechanical characterization included Vickers hardness HV10 (98.1 N force, 5 indents each, Bareiss, Oberdischingen, Germany), measurement of Young's modulus E by acoustic emission technique (IMCE, Genk, Belgium) and a measurement of bending strength $\sigma_{4pt}$ and fracture resistance $K_{ISB}$. For the bending tests bars of ~2 × 4 mm$^2$ cross-section, >25 mm length was tested in a 4pt setup with 20 mm outer and 10 mm inner span (polished side on tensile side, 10 bars each, crosshead speed 0.5 mm/min, Zwick, Ulm, Germany). For toughness measurements, test bars were indented with a HV10 indent in the middle of the polished side with cracks oriented parallel and perpendicular to the sides. The bars were placed with the indent on the tensile side within the inner span, and the residual strength was measured (same setup, crosshead speed 2.5 mm/min). Fracture resistance $K_{ISB}$ (indentation strength in bending) was calculated from measured values of residual strength, hardness and Young's modulus according to Chantikul [31].

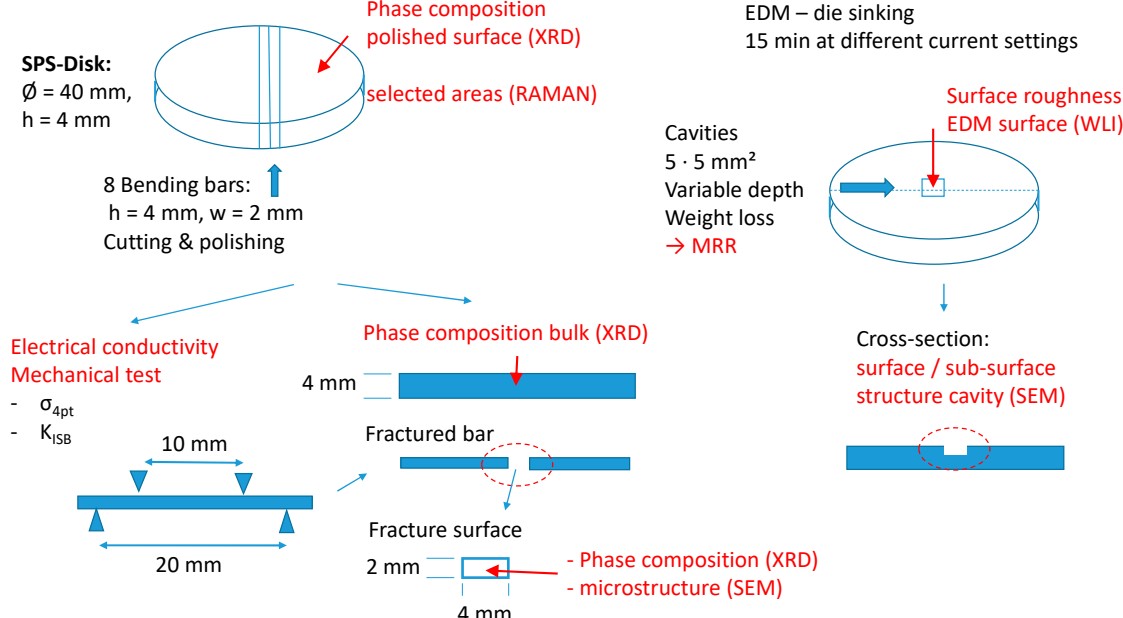

**Figure 1.** Overview of sampling and testing scheme.

The electrical conductivity of the materials was measured with a self-built 4 pt measurement device. Electric discharge machining experiments were carried out by die sinking of rectangular cavities of 5 × 5 mm$^2$ (15 min, 2 repetitions) using copper electrodes that were previously ground and polished to the machining experiments (AEG Elbomat, Remscheid, Germany, oil based dielectric). The material removal rate was determined gravimetrically.

$$MRR = V/t \ [mm^3/min], \ with \ V = (m_t - m_{t=0})/\rho \ [g/(g/mm^3)]$$

Machining tests were carried out using three different current settings. The discharge gap was set automatically by the machine. As the machine only allows setting fixed machine parameters, real discharge currents and pulse durations were measured with a current probe attached to an oscilloscope (Table 1). At identical machine settings, the measured

values differed slightly for three different materials. Discharge currents increase with tungsten carbide content while discharge time declines, and this effect is more pronounced in machine settings with higher discharge energies.

**Table 1.** Measured discharge currents and pulse durations for different machine settings and TZP-WC-materials.

| Machine Setting | Discharge Current (A) [1] | Pulse Duration (μs) [1] |
|---|---|---|
| CS1 | 3.9; 4.05; 4.2 | 3.7; 3.7; 3.65 |
| CS2 | 7.15; 7.85; 8.35 | 5.1; 4.6; 4.4 |
| CS3 | 10.25; 10.25; 11.65 | 6.75; 5.5; 5.25 |

[1] Values listed are given sequentially for 24WC; 28WC; 32WC.

The surface roughness of the machined materials was determined by white light interferometry (WLI) (Bruker, Karlsruhe, Germany) to determine surface roughness. Top views of machined cavities and polished cross-sections through machined samples were studied by SEM (Hitachi S800, secondary electrons, 5 kV acceleration voltage). The microstructure of fracture surfaces of ISB bars was studied by HR-SEM (FEI Helios Nanolab 600, in lens secondary electrons, 5 kV acceleration voltage, current 5.4 pA). The phase composition of zirconia was determined by XRD (X'Pert MPD, Panalytical, NL, CuK$_{\alpha 1}$, Ge-monochromator, Bragg-Brentano setup, accelerator detector) according to the calibration curve of Toraya [32].

## 3. Results

### 3.1. Mechanical Properties

Figure 2a shows the Vickers hardness HV10 and Young's modulus E of TZP-WC composites depending on WC fraction. Both hardness and Young's modulus show a linear increase with increasing tungsten carbide content as we may expect by the rule of mixture as tungsten carbide has higher hardness and Young's modulus than zirconia. The bending strength and fracture toughness of the materials are plotted in Figure 2b.

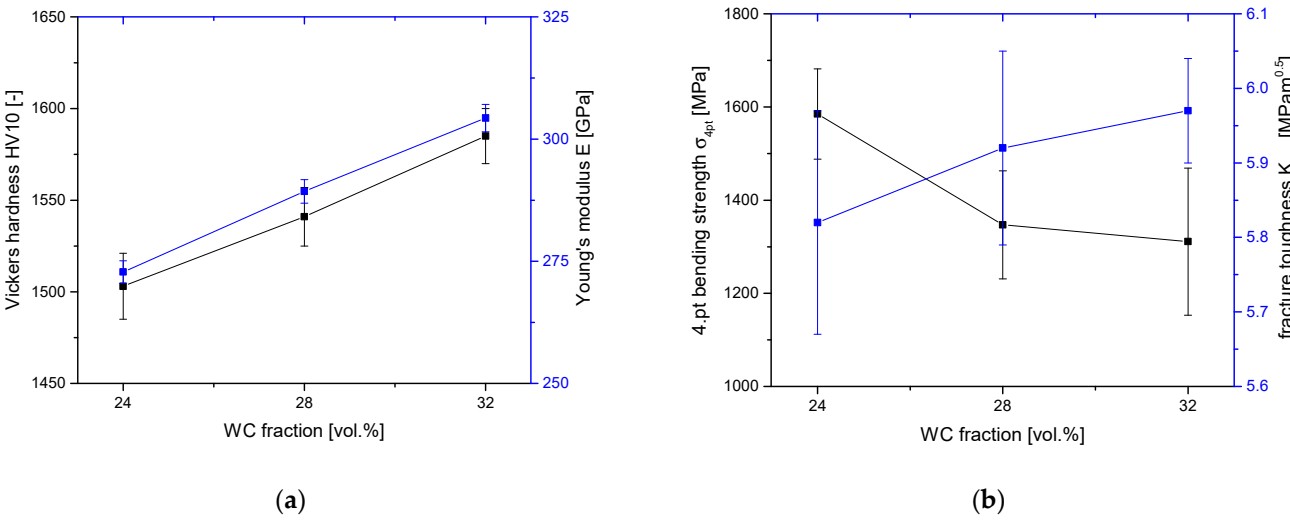

(**a**)  (**b**)

**Figure 2.** (**a**) Vickers hardness HV10 and Young's modulus E of TZP-WC composites depending on WC fraction. (**b**) Bending strength $\sigma_{4pt}$ and fracture toughness K$_{ISB}$ of TZP-WC composites depending on WC fraction.

Bending strength declines from 1600 MPa in 24 WC to a level of 1300–1350 MPa for materials with higher WC content. The fracture toughness of the materials is in the range between K$_{ISB}$ = 5.8 and 5.95 MPa·$\sqrt{\text{m}}$. Toughness shows an adverse trend to strength; the differences in toughness are, however, quite low and hardly exceed the standard

deviation. The density of the samples (not shown in detail) ranged between 99.5 and 99.9% of theoretical. The complete set of mechanical property data is provided in Table S1: Supplement A_Mechanical properties.

### 3.2. Electrical Conductivity

Electrical conductivity of the composites is shown in Figure 3. In the observed range, conductivity increases linearly with increasing WC content from 10 kS/m at 24 vol.% WC to 68 kS/m at 32 vol.% WC. In the vicinity of the percolation limit, however, an exponential increase in conductivity would be expected [33].

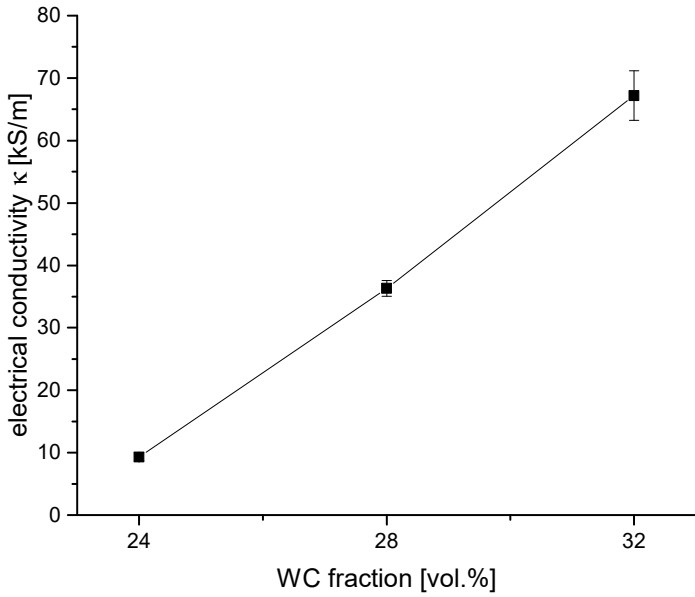

**Figure 3.** Electrical conductivity of TZP-WC composites depending on WC fraction.

### 3.3. Microstructure

An overview of the microstructure of fracture surfaces is shown in Figure 4a–c. The microstructure of all samples is obviously very fine grained. 24WC shows dense and very homogeneous structure, and only a few black spots indicate some irregularities. The fracture mode is mixed inter- and transgranular. In 28WC, these black regions become more frequent but stay in the size range of ~1 μm. In the case of 32WC, these dark regions become more frequent and much larger and are in some cases connected. The detailed images in Figure 5a–c show some more details. While the microstructure outside the dark regions is more or less comparable in all samples, the average grain size is <200 nm for both TZP and WC, and the intergranularly fractured grains exhibit sharp edges. The grains with transgranular fracture exhibit roughness in the nanoscale. In the dark regions, however, the grain structure of the material below is covered with an amorphous layer. The fracture proceeds through this layer without exhibiting structural features. The high contrast to the surrounding microstructure indicates a compositional difference. The dark color indicates the presence of a compound with a lower average atomic number, presumably carbon.

### 3.4. Raman-Spectroscopy

Larger lamellar regions—corresponding to the dark regions in the fracture surfaces—were identified by optical microscopy in the surface of polished parts. It was a priori assumed that carbon—possibly graphene—was formed during the SPS process by thermal decomposition of the tungsten carbide. These lamellae were, therefore, studied by Raman-spectroscopy in order to be able to identify the assumed carbon precipitates. Figure 6a shows a Raman spectrum of a lamella in 24WC.

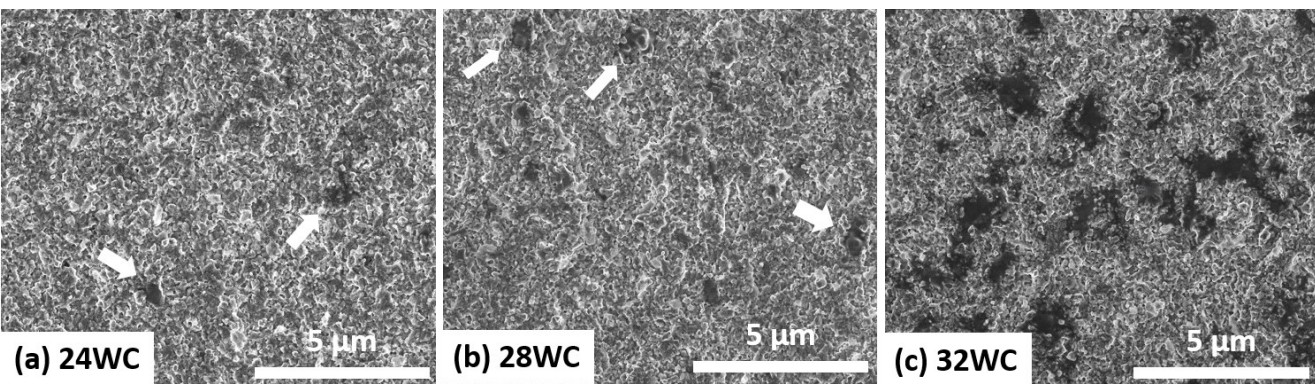

**Figure 4.** Overview SEM images of fracture surfaces of samples broken in ISB test. Black regions in TZP-24WC and TZP-28WC indicated by white arrows: (**a**) TZP-24WC, (**b**) TZP-28WC and (**c**) TZP-32WC.

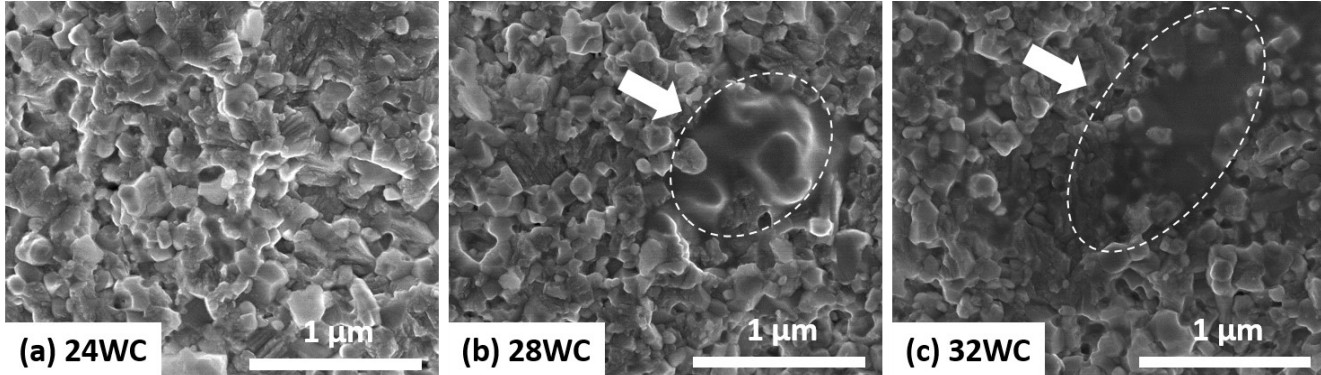

**Figure 5.** Detailed SEM images of fracture surfaces of samples broken in ISB test. Dark regions with amorphous structure marked with arrows dotted ellipses. (**a**) TZP-24WC, (**b**) TZP-28WC and (**c**) TZP-32-WC.

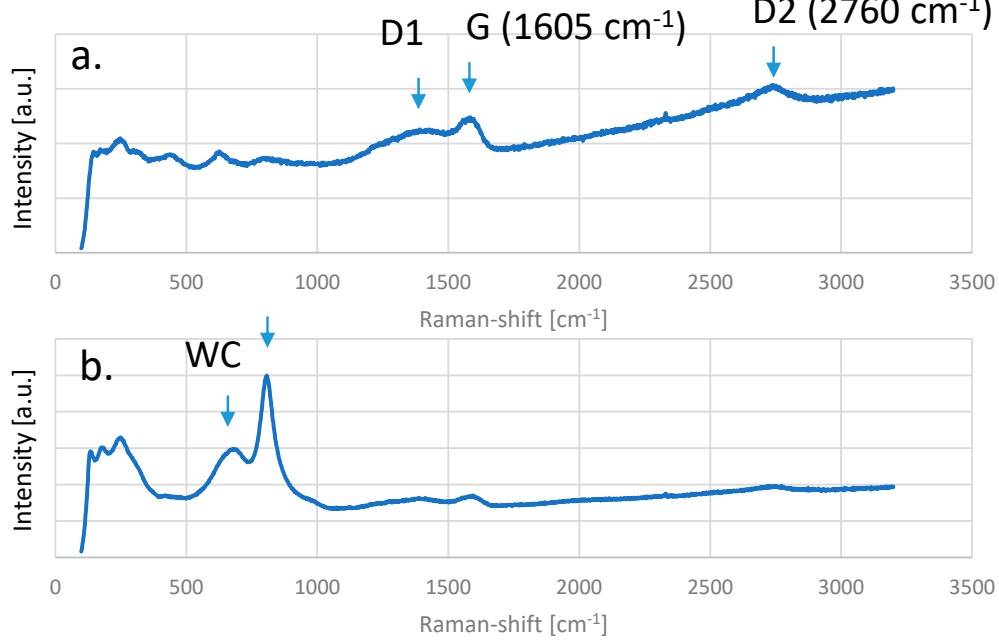

**Figure 6.** Raman spectra of TZP-24WC: (**a**) pristine material and (**b**) repeated measurement at the same location.

Three typical bands 1D, 2D and G for graphene are visible. The bands are, however, not very sharp. The presence of the broad d-band indicates that the graphene structure is either highly defective or that it is partly amorphous. The location of the G band typically can provide some hints for the number of graphene layers as the Raman shift according to Wang is correlated to the number of layers ($\omega_G = 1581.6 + 11/(1 + n^{1.6})$) [34]. The position of the G peak at 1605 cm$^{-1}$ is out of the reasonable range (1581.6 cm$^{-1}$ for Graphite, 1587 cm$^{-1}$ for monolayer graphene; this would correspond to negative numbers of layers). It may, however, be assumed that the graphene, as it is not a free particle but embedded in the composite, is under compressive residual stress, which leads to a shift to higher wavenumbers [35]. A repetition of the measurement at the same place is not possible as the majority of the carbon inclusion is evaporated by laser irradiation. The remaining signals are very weak, the relict (Figure 6b) shows the predominant bands of tungsten carbide at 710 cm$^{-1}$ and 810 cm$^{-1}$. Spectra measured for 28WC and 32WC materials (not shown) were similar.

### 3.5. Phase Composition

The phase composition of the zirconia matrix is predominantly tetragonal; monoclinic contents ranged between 2 and 5%. An exact quantification is difficult due to the fact that the large 001 peak of WC coincides with the 111 peak of monoclinic zirconia. Figure 7 shows XRD spectra of the cross section (bulk material) of TZP-WC composites in the 2θ-range of 27–55° (measurement made in the center of the broad side of a cross cut bar, see Figure 1). Peaks of tungsten carbide WC, tetragonal zirconia, ditungsten monocarbide W$_2$C and tungsten W can be identified. Bulk TZP-WC contains W$_2$C in all cases; moreover, TZP-28WC contains a small amount of tungsten.

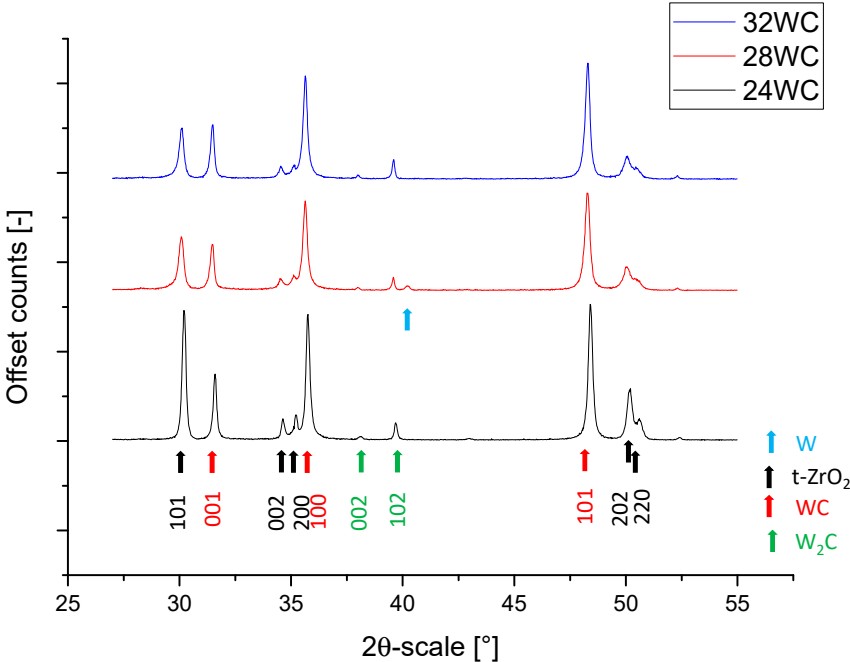

**Figure 7.** XRD patterns of TZP-WC, cross-cut section and bulk material.

Figure 8 shows the XRD spectra of the polished surface of TZP-WC composite disks (top view onto the polished surface of the entire sintered disk, see Figure 1). The polished surfaces show only tetragonal zirconia and tungsten carbide. Peaks of W$_2$C at 34.5° and 39.5° 2θ theta are absent.

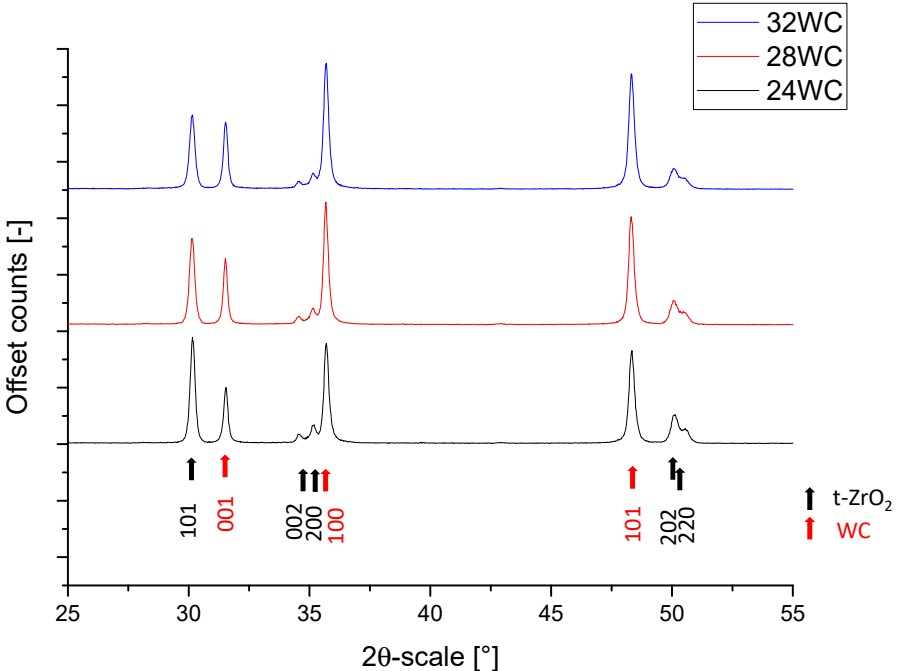

**Figure 8.** XRD patterns of TZP-WC polished surface of sintered disk.

### 3.6. Electric Discharge Machining

Figure 9a shows the material removal rate (MRR) depending on tungsten carbide fraction and machining parameters (the discharge current increases from setting 1 to 3). Two tendencies are evident. MRR increases with increasing discharge current. This observation is quite trivial as it shows discharge energy and, thereby, the locally dissipated heat scales with the discharge current. The second effect is related to the tungsten carbide content. At the lowest current setting CS1, machining speed is independent of WC current. At higher discharge currents, MRR decreases with increasing WC content. The effect is even more pronounced at the highest current setting. The second important issue is the resulting surface quality. Figure 9b summarizes the results of surface characterization by white light interferometry.

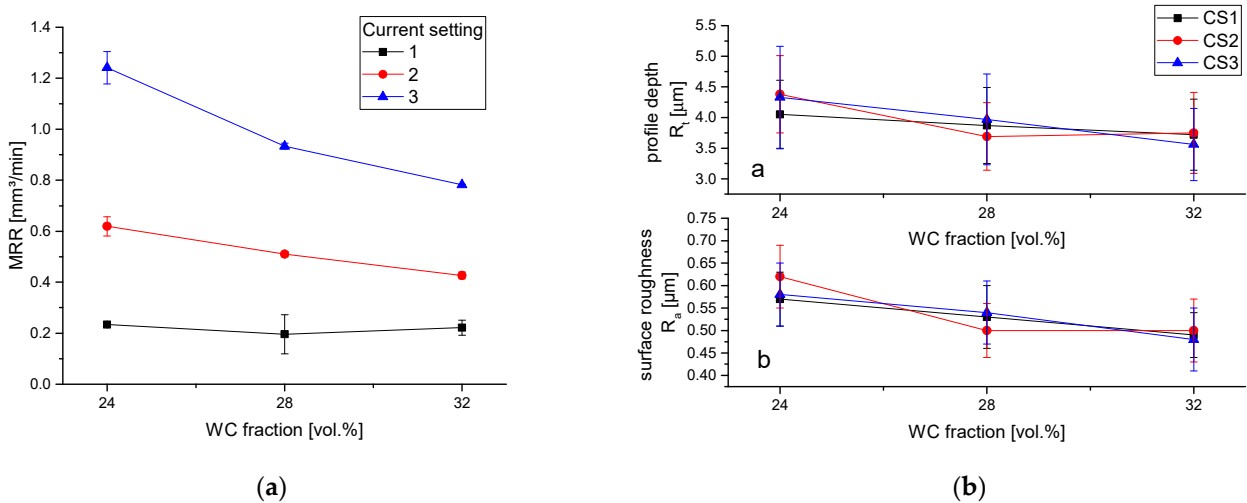

(**a**)  (**b**)

**Figure 9.** (**a**) Material removal rate MRR depending on current settings and WC fraction. (**b**) Surface characteristics determined by WLI depending on current settings and WC fraction; (**a**) profile depth $R_t$; (**b**) surface roughness $R_a$.

The effect of the current setting on surface roughness $R_a$ and profile depth $R_t$ is moderate. The dominant parameter is the WC content. The higher the WC content, the smoother the surfaces and the lower the profile depth.

A detailed view of the structure of the machined surfaces is obtained by white light interferometry. Figure 10 shows the structure of 24WC and 32WC materials machined with three different current settings (Table 1).

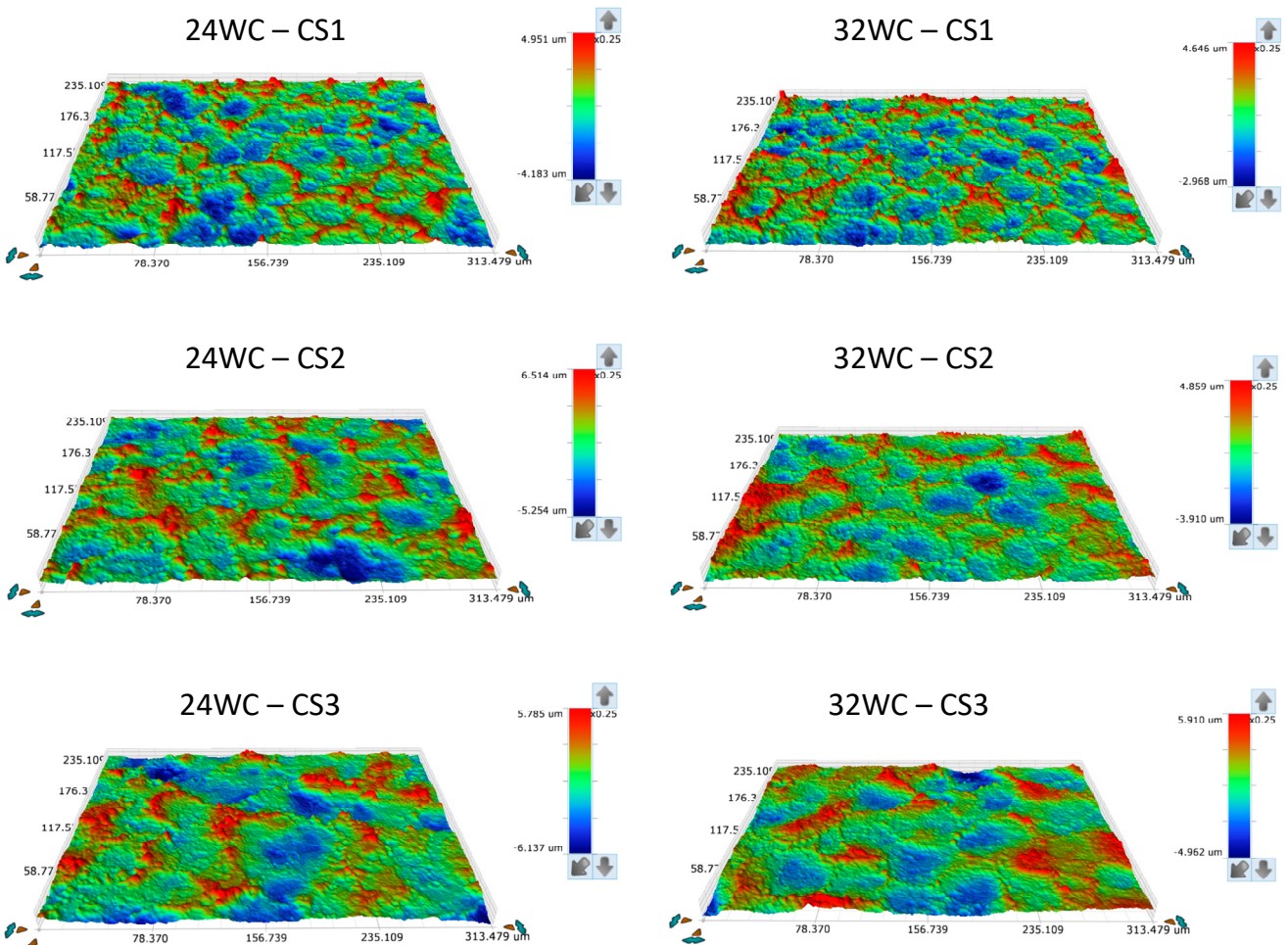

**Figure 10.** Surface structure in ED-machined 24WC and 32WC materials machined with current settings CS1, CS2 and CS3 determined by white light interferometry' different colors mark depth profile (scale bar at the side of each image).

As material removal is created by a multitude of single discharges, the surface exhibits a large number of overlapping circular regions with an elevated rim and a depressed center region. The size of these discharge craters increases with the energy input as expected. Moreover, it is clearly visible that the materials containing higher fractions of WC are characterized by average smaller discharge craters, a more regular discharge pattern and surface structure and a flatter profile, which visually confirms the above-mentioned integral roughness values.

Figure 11 shows the average size of discharge craters depending on WC content and machine settings. The statements made above are confirmed and quantified. The discharge crater sizes increase with discharge energy and decrease with WC content of the composites. The largest difference is observed between 24 and 28 vol.% WC. The difference between 28 and 32 vol.% is less pronounced.

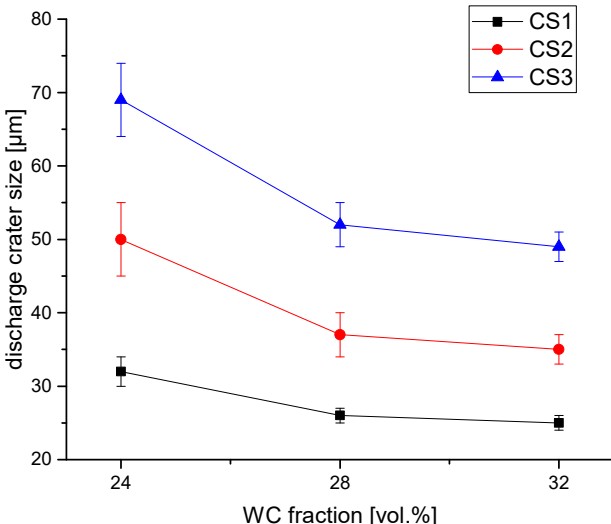

**Figure 11.** Average size of discharge craters in TZP-WC depending on WC fraction and current setting; determined from SEM images.

More detailed data on the surface properties and EDM results of the machined materials are available in Tables S2: Supplement B_surface properties and S3: Supplement C_EDM results.

Cross sections pf the machined surfaces enable an assessment of the subsurface damage triggered by the EDM-process. Figure 12 shows the subsurface damage of TZP-WC materials depending on the type of material and machine settings. One obvious observation for the cross sections is that subsurface damage increases with discharge energy irrespective of material composition. The cracks formed under CS1 and CS2 conditions are oriented at an angle of 60–90° to the surface. The length of these cracks hardly exceeds 5 μm. In some cracks, a bifurcation is observed. Some samples show the remnants of a layer at the surface, which contain large globular WC inclusions in a WC depleted matrix. Evidently, the material has melted during the EDM process. In the liquid, melt zirconia and WC have separated. If the layer is not completely removed, these remnants resolidify on the surface.

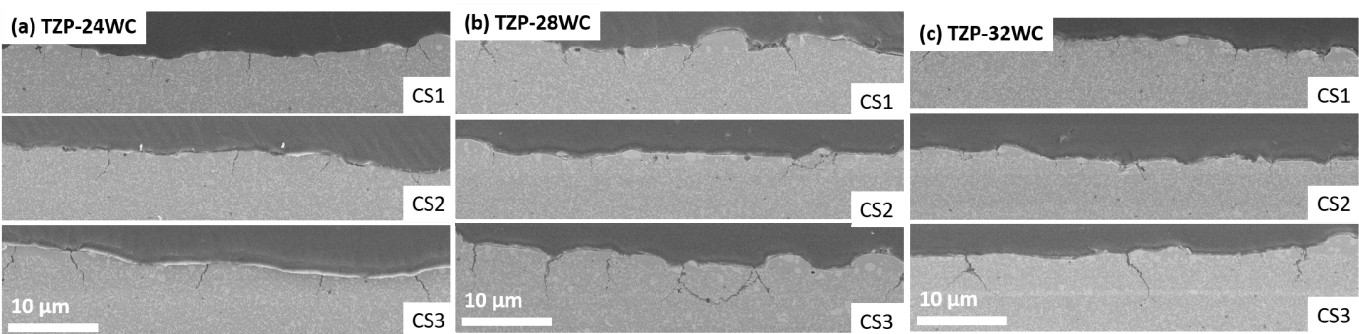

**Figure 12.** SEM images of cross sections through the machined surfaces; (**a**) TZP-24WC; (**b**) TZP-28WC; (**c**) TZP-32WC.

Samples machined with parameter set CS3 show larger cracks. These cracks show a larger crack opening, and as they proceed to the bulk, they change their direction. Single cracks may extend to a length of 8 μm. Some cracks merge and embrace material, which is then prone to spall off (24WC, CS3). Except for some smaller regions where resolidified material is present, the surfaces are very smooth, and the processing-affected zone along the surface is very narrow.

Some interesting further details are visible in SEM images of machined surfaces (Figure 13). The surfaces show an extremely smooth glass-like surface interrupted by

entrapped gas bubbles and shrinkage cracks induced by resolidification. The serrated morphology of the cracks shows that the resolidified layer is not amorphous but nanocrystalline. This thin resolidified layer may also spall off, exhibiting the surface of the bulk material below.

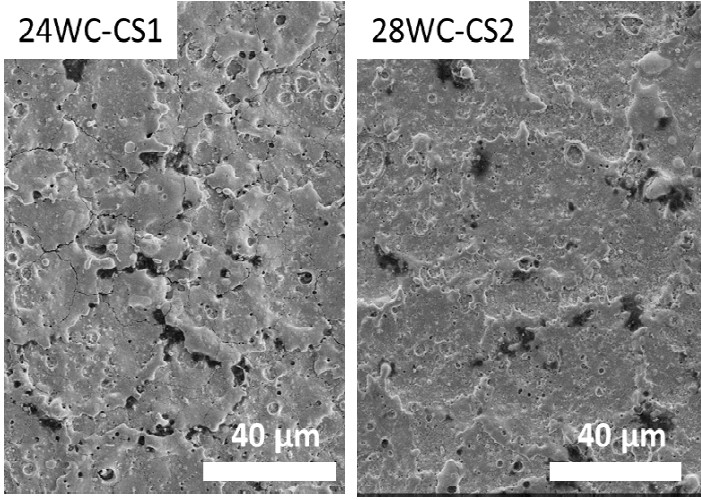

**Figure 13.** Black spots in TZP-WC machined with CS1.

Samples with 24–28% WC machined with CS1 all show small black spots, especially around discharge craters. A close look at these black spots shows that, in the case of 24WC, a part of the spots is caused by topography effects (open bubbles). In case of 28WC, these black spots are related to subsurface features (light elements such as carbon) with different interaction with electrons. (Figure 13). The dark spots completely disappear at higher WC content (32% WC) and at high current settings (CS2 and CS3).

Another interesting detail is the occurrence of fibrous structures on the surface of machined materials (Figure 14). These fibers appear very frequently, especially on the surfaces of 32WC material irrespective of current settings. The fibers are typically 5–15 μm long with diameters typically well below 1 μm. Such fibrous structures were reported also recently by Gallardo for TZP-graphene ED-machined at high current settings and interpreted as carbon nanofibers [36]. As these fibers appear very specifically in TZP-32WC, it may be speculated that they are not formed by decomposition of oil-based dielectric but are decomposition products of the ceramic material.

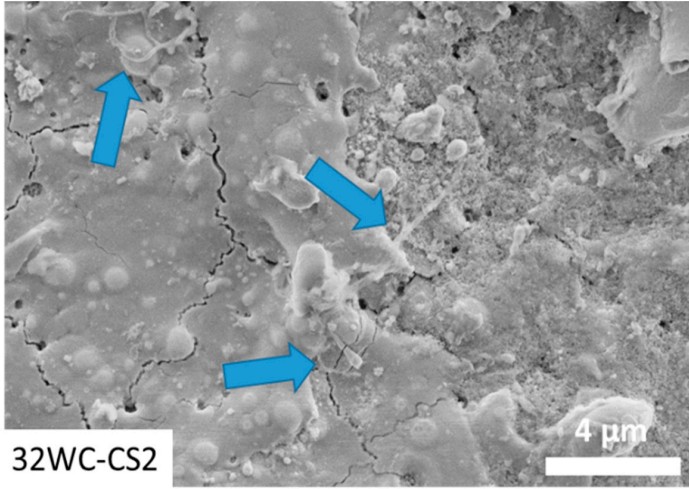

**Figure 14.** Fibrous structures on top of ED-machined surface of TZP-32WC machined with current setting CS2; blue arrows indicate fibers.

## 4. Discussion

The results in this study on SPS sintered 1.5Yb1.5Sm-TZP-WC materials show that such materials can be consolidated to full density without difficulty at short process times. The mechanical properties are attractive and suitable for mechanically demanding applications. From a scientific point of view, the observed "side-effects" are, however, much more interesting. Fortunately, a study on the same materials consolidated by hot pressing is available so that the differences between hot pressing and SPS can be analyzed [30]. In the case of the hot pressed material, strength increased slightly with WC content (1550–1700 MPa at a sintering temperature of 1400 °C) and toughness on average slightly declined with rising WC content. In cases of SPS, the effects are observed to be vice versa. Hot pressed materials were harder (up to 1670 HV10 for 32WC) than spark plasma-sintered materials at identical density values.

In the present case, TZP-WC materials showed an increasing amount of carbon precipitates in the microstructure with increasing tungsten carbide content. It seems that the free carbon in the microstructure which was identified by Raman spectroscopy results in a loss of strength and hardness and a slight increase in fracture resistance.

XRD analysis shows occurrence of $W_2C$ in the bulk of all TZP-WC materials. Tungsten carbide decomposing to $W_2C$ seems to be the most probable source of the carbon (Equation (1)).

$$2WC \rightarrow W_2C + C \tag{1}$$

However, such carbon precipitates were not found in hot pressed material. In hot pressed material, free carbon—if any—is probably formed very slowly and homogeneously all over the ceramic volume and under more or less isothermal conditions. The longer dwell of 2 h facilitates reaction, e.g., with $ZrO_2$ to form volatile CO and substoichiometric zirconia (Equation (2), not stoichiometrically equilibrated).

$$C + ZrO_2 \rightarrow ZrO_{2-x} + CO \uparrow \tag{2}$$

Haberko et al. [37] proposed that, at ambient pressure, a reaction of Zirconia with WC takes place, which results in the formation of zirconium carbide, ditungsten carbide and CO (Equation (3)), provided that the CO partial pressure is low enough (<0.95 atm at 1400 °C, <5.2 atm at 1500 °C).

$$ZrO_2 + 6WC \rightarrow 3W_2C + ZrC + 2CO \tag{3}$$

Zirconium carbide formation was, however, not observed, which could be due to the fact that, in the almost dense part in a mold at a pressure of 60 MPa, the CO pressure may exceed this value. An incorporation into the anionic sublattice of zirconia in low amounts could also be possible [38]. In cases of SPS, the mechanism seems to be different. Carbon evidently forms localized deposits. Fracture through these carbon deposits shows a featureless fracture surface, and the results of Raman spectroscopy show a spectrum containing the typical pattern of graphene. The broad peaks, however, indicate carbon, which is only partially crystallized.

It may be speculated that carbon formation and aggregation processes are caused by the interaction of the two phase composite powder mixture with the applied pulsed direct current. At the beginning of the sintering cycle, the powder mixtures (which are designed to be conductive as fully densified ceramics) are non-conductive. Consequently, initially, the current flows through the graphite die and not through the sample. The sample is indirectly heated by conduction from hot die walls. As the powder mixture is heated up progressively, densification occurs and the electrically conductive phase finally forms a percolating network. Now, the current starts flowing through the sample or—to be more precise—through the percolating WC network. The WC network acting as a thermal resistor is, therefore, initially severely overheated and may decompose. Literature on this topic is rare. Girardini reported on formation of $W_2C$ from WC and surface tungsten oxides in

consolidation of nanoscale binderless WC by SPS. However, in this case, no free carbon was observed; on the contrary, carbon doping prevented $W_2C$ formation [15].

Nayak reported on eutectoid decomposition of tungsten carbide by thermal arc plasma treatment. They confirmed formation of $W_2C$, free carbon and elementary tungsten under such conditions. (T > 2500 °C) [39]. If this holds true, locally, the temperature is probably much higher than measured 10 mm above the sample surface by the pyrometer. The surrounding nanoscale TZP matrix, especially if it is still porous, is a good thermal insulator. Thermodynamically, $W_2C$ formation may already start above 1525 K [40]. The aggregation of carbon, however, implies much higher temperatures in order to form carbon at much higher rates than typical carbon-consuming reactions (reaction with zirconia and formation of CO, Equation (2)). Possibly close to the punches where no $W_2C$ formation is observed (Figure 8, polished surface of SPS disk), the temperature level during sintering is significantly lower than in the sample center. Two effects may account for this: the locally higher densification at the punches (typical for axial pressing), which leads to higher densification and lower thermal resistance at an earlier stage; and the vicinity of the graphite die acting as a thermal drain. The larger carbon deposits in composites with higher WC fraction support this interpretation. The higher the WC content, the more difficult densification is and the more probable the formation of a rigid WC scaffold in the intermediate sintering stage, which favors local overheating and the decomposition of WC. By XRD, $W_2C$ is only found in the center of the samples (Figure 7, measured in the middle of polished bending bars which were cut upright from the initial disks, see Figure 1).

From a viewpoint of machining the carbon, which is conductive, it seems to have little influence, and the EDM process works regularly. Machining speed and surface quality are not negatively affected. In detail, some interesting effects were observed, such as the occurrence of dark spots in the subsurface region of 24WC and 28WC at low discharge energies. At larger discharge energy, these features are obviously destroyed (evaporated) by the influence of the electric discharges. In cases of ED-machining of WC/Co in water, WC is split into tungsten and carbon. Both are then oxidized by oxygen containing dielectric, releasing carbon dioxide and solid debris of tungsten and tungsten trioxide [41]. In the hydrocarbon dielectric used in this study, oxidation is apparently prevented and free carbon may occur unless the discharge energy is sufficient to completely evaporate it. Fibrous material occurring predominantly in 32WC (much less pronounced in other materials) indicates that decomposition products form some kind of carbon nanofibers if exposed to high discharge currents. The formation of finer discharge patterns on WC-rich composites is straightforward as a higher electrical and thermal conductivity results in a lower local Joule heat. The opposite effect is responsible for the higher material removal rate in composites containing less WC.

Finally, many questions remain open, and more detailed investigations are required to explain the observed effects and falsify tentative explanations.

Concerning future studies, two possible goals can be identified. Assuming that carbon formation is undesired and that it results in inferior mechanical properties, to changing SPS process parameters in such a way as to avoid carbon formation in general or at least its uneven distribution in larger deposits is suggested. In the case of SPS blanks produced at technical scales for mechanical engineering applications (~20 cm diameter, ~5 cm height), which have a much higher volume to surface ratio, it can be expected that overheating will be even more severe.

The second approach could be to try and deliberately produce ceramic–carbon (if possible graphene) composites by in situ graphene formation during SPS. Composites with graphene addition by a mixing process suffer from texturing and agglomeration. An in situ method could help deposit graphene homogeneously throughout the microstructure and, therefore, fully exploit its reinforcement/toughening effect.

## 5. Conclusions

1.5Yb1.5Sm-TZP composites with 24–32 vol.% electrically conductive WC dispersion were successfully densified by SPS to full density. The materials show an attractive combination of mechanical properties and fine grain microstructure. The composites exhibit high electrical conductivity, which enables ED-machining by die sinking at high rates and produces high surface quality. The zone of subsurface damage caused by EDM is small. Its size scales with discharge energy. Composites with high WC exhibit lower surface roughness but at a reduced machining speed in EDM.

Evidently, the composites contain locally aggregated free carbon deposits. The size of carbon deposits scales with WC content, and this is probably caused by thermal decomposition of WC forming $W_2C$ and carbon in locally overheated regions during SPS. The formation of free carbon may explain the differences in mechanical properties compared to hot-pressed materials of the same composition.

**Supplementary Materials:** The following supporting information can be downloaded at: https://www.mdpi.com/article/10.3390/jmmp6020028/s1, Table S1: Supplement A_Mechanical properties, Table S2: Supplement B_surface properties, Table S3: Supplement C_EDM results.

**Author Contributions:** Conceptualization, F.K.; methodology, F.K.; investigation, E.W., F.K. and M.R.; data curation E.W.; writing—original draft preparation, F.K.; writing—review and editing, F.K.; visualization, E.W. and F.K.; supervision, F.K.; project administration, F.K.; funding acquisition, F.K. All authors have read and agreed to the published version of the manuscript.

**Funding:** The authors gratefully acknowledge the funding of the project by Deutsche Forschungsgemeinschaft (DFG), Germany, grant number KE879/3-3.

**Data Availability Statement:** Data are contained within the Supplementary Materials for the article.

**Acknowledgments:** Lauxman F. and Berthold C., Competence Center Archaeometry-Baden-Wuerttemberg (CCA-BW), University of Tübingen, for Raman spectroscopy and W. Schwan, IFKB, University of Stuttgart, for optical microscopy and SEM images.

**Conflicts of Interest:** The authors declare no conflict of interest. The funders had no role in the design of the study; in the collection, analyses or interpretation of data; in the writing of the manuscript; or in the decision to publish the results.

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
