# Peer review of "Spark Plasma Sintering of Electric Discharge Machinable 1.5Yb-1.5Sm-TZP-WC Composites"

_jmmp, doi:10.3390/jmmp6020028_

Round 1

Reviewer 1 Report

The paper presents value for both science and industrial applications. Some remarks are listed below:

  1. It would be better to present a scheme of the sample for the bending test.
  2. Figure 10: on the y-axis, there is a misspelling in the word “discharge”
  3. In the discussion, the formation of carbon and considerations on mechanism should be supported by references.

Author Response

Dear Reviewer 1:

Thank you very much for your valuable comments, we hope that all open questions are adressed in the revised manuscript (changed passages marked in red in the revised manuscript):

In detail:

  1. It would be better to present a scheme of the sample for the bending test.

This seemed to be also a problem with the comments of the other reviewers, we evidently did not describe the procedures sufficiently.

  • For a better understanding of how samples were prepared, cut and machined and how and where which measurements were made we decided to add an overview of the sampling and testing schemes (new fig. 1).
  1. Figure 10: on the y-axis, there is a misspelling in the word “discharge”

Was corrected

  1. In the discussion, the formation of carbon and considerations on mechanism should be supported by references.

Some equations and references are added, additional literature was studied and cited

In fact we could find some further indications in literature that our interpretation is correct,

Reviewer 2 Report

This is an interesting article and the manuscript was written in decent English.

1- In the introduction, it is recommended to refer to some SPS consolidation of metallic materials as well.

(a) Microstructure globularization of high oxygen concentration dual-phase extruded Ti alloys via powder metallurgy route

(b) Improved ductility of spark plasma sintered aluminium-carbon nanotube composite through the addition of titanium carbide microparticles

(c) Ultrafine-grain formation and improved mechanical properties of novel extruded Ti-Fe-W alloys with complete solid solution of tungsten

2- How do you confirm the amorphous layer in the dark regions of Fig. 4? And, also how do you verify the compositional difference in these regions? What was the SEM captured mode, BSE or SE?

3- Why tungsten peak was only appeared in 28WC and what was your intention to show two types of XRD profiles in Fig. 6 and 7? XRD normally is performed on polished surfaces and please explain why you have shown XRD profile in Fig 6?

Author Response

dear Reviewer 2:

thank you for your valuable comments and suggestions, we hope that we have explained all open questions satisfactorily, the revised manuscript contains changed text-passages in red.

  1. Additional literature on the use of SPS in different fields was added.
  2. Such flake-like deposits were analysed by RAMAN and identified as carbon. The microstructure of the ceramic visible below the black spot has rounded shape while the fracture surface material in the surrounding unaffected material shows well crystallized. As the crack split the dark region and there is definitively an easy to cleave separate phase which separates the microstructure below and above the carbon deposit. Such rounded/flattened surfaces were also found in a recent paper in contact area between graphene and zirconia (17. Obradović, N. Kern, F. Properties of 3Y-TZP zirconia ceramics with graphene addition obtained by spark plasma sintering, Ceram. Int. 2018, 44, 16931-16936.) In fact we must admit that further studies are required to prove this presumptive evidence.
  3. This is now made much clearer by addition of a a sampling scheme (new Fig. 1) which shows which samples were taken in which position and which analytic methods were used. The SPs disk was polished and Figure 8 represents the composition at the outer surface of the disk (which was in close contact with the die and punches. Here no W2C, and no tungsten is observed.

Figure 7 shows the composition of the material in the sample center, we cut a bending bar upright and checked the composition in the middle of the bending bar by XRD, this is the bulk composition which clearly shows W2C and in one case also tungsten.

Our argumentation in the discussion is the following. Close to the die the heat exchange is sufficient to prevent overheating of the sample, the surface conditions are the conditions we measure by the pyrometer (regular sintering temperature). In the bulk an overheating occurs as at a certain point the WC dispersion becomes percolating and forms a rigid scaffold which conducts the electric current. As the conductivity of this scaffold is initially relatively low the resistive heating effect is high. The WC scaffold overheats so that decomposition to W2C and carbon may occur. The higher the WC content the earlier this overheating effect occurs and the more rigid is the WC scaffold. In case of low WC content the scaffold is less rigid and the densification of the composites is faster.

Reviewer 3 Report

The paper deals with a popular topic of ceramic matrix composite with enhanced electrical conductivity, in order to achieve easier machinability by EDM. It is well writen organized, the language is correct, it is logical and reads well.
It is sort of a continutation of some previous works. FOr this reason, some useful information is omitted, particularly concerning the detail of the materials. Yet, I believe, since that info is in another journal (Ceramics) some of it should be repeated also hereand also some more explanations should be made.
Recommendations and comments:
- At the end of the Introduction, the AIM of work should be clearly spelled. Otherwise, the reader know what is done, but it is unclear why.
- Please, provide also some info, why such composition. What is the reason for the presence of Yb and Sm.
- The amount of WC is rather high (up to 32%). Is it not some effort to minimize this content? Or conversely, would it not be better to simply go for WC? It is conductive, very hard, reasonably tough and strong. With the exception of 4-point bending strength, (and possibly the speed of EDM cutting) everything improves with more WC. It comes back to the question about the aim of work.
- The dark regions in Fig. 3 are carbon rich deposits? It is not clear to me. It could be shown better. They are presumably artifacts due to SPS. Do they form in HP?

To summarize, the paper is sound, it should provide a little more info and some explanations.   

Author Response

dear Reviewer 3:

Thank you very much for your remarks. You adressed some critical points indeed. The aim of the study was actually a very simple one to find out if SPS changes something substantial in properties and ED-machinability of thes materials. The story with the carbon deposits came up somewhat unexpected but may be even more interesting than the rest, we will try to adress this topic in future studies with partners who have the necessary analytical equipment. We tried to clarify the aim of the study and explain a bit more on the materials used (at first sight a bit exotic stabilizer composition) and the applications targeted. We hope that we have answered the questions satisfactorily.

  • At the end of the Introduction, the AIM of work should be clearly spelled. Otherwise, the reader know what is done, but it is unclear why.

1.5Yb-1.5Sm-TZP-WC composites based on identical powder mixtures were previously manufactured by hot pressing and characterized with respect to ED-machinability me-chanical and electrical properties, ED-machinability and microstructure [23]. In the present study exactly the same recipes were densified by SPS in order to check if materials with identical properties and EDM characteristics are obtained

Please, provide also some info, why such composition. What is the reason for the presence of Yb and Sm.

You are right the composition is a bit unusual. A comment was given and an additional reference.

The stabilizer recipe of the zirconia matrix has a strong influence on the transformability of the tetragonal phase, the achievable transformation toughness and thereby the me-chanical properties. Hence it can be used to tailor the mechanical properties of TZP-WC composites. The critical parameters are the ionic radii of the stabilizer cations, their con-centration and their distribution. 1.5Yb-1.5Sm costabilized TZP is therefore a compromise between the state of the art 3Y-TZP which leads to composites with high strength but lim-ited toughness and 1.5Y-1.5Nd-TZP based materials which offer ultimate toughness but limited strength [Gom21]. 

- The amount of WC is rather high (up to 32%). Is it not some effort to minimize this content? Or conversely, would it not be better to simply go for WC? It is conductive, very hard, reasonably tough and strong. With the exception of 4-point bending strength, (and possibly the speed of EDM cutting) everything improves with more WC. It comes back to the question about the aim of work.

To elucidate the optimum composition was the aim of a joint research project (see funding) with RWTH Aachen Germany, we made the composites they did the EDM tests.

In fact TZP-WC can be machined at even lower WC content but the high energy input especially during wire cutting leads to cracks which are very large and are a result of thermal shock, the process becomes instable and the machining result is not satisfactory. We could die sink materials with as little as 12 vol% WC (unpublished). But the electrical conductivity is not the only precondition. ZrO2-rich materials have a large CTE, a low thermal conductivity and so all preconditions for thermal shock are fulfilled. The WC has high thermal conductivity and low CTE. So typically ED-machinable TZP materials have WC or TiN contents in the range of 30-40%, far more than required to fulfil the requirements to electrical conductivity.

Pure WC is hard extremely heavy but has much lower toughness. Moreover it requires much higher sintering temperatures. Another point is that the applications are typically tool inserts in steel mold, dies or tools, e.g for injection molding of metals ceramics or highly abrasive Glass fiber filled polymers which wear down steel molds very quickly. Therefore the material has to show a low CTE-mismatch to steel. So that there is no gap or stress concentration between the two materials upon thermal cycling.

 So coming back to the aim we added:

As EDM is a contactless and force-free non-conventional machining technology capable to machine materials irrespective of high hardness and abrasion resistance and is therefore a very attractive technology for ceramic materials [1]. Ceramics are also interesting candidates for such applications due to their high hardness and chemical stability.  

- The dark regions in Fig. 3 are carbon rich deposits? It is not clear to me. It could be shown better. They are presumably artifacts due to SPS. Do they form in HP?

Such deposits were never found in materials made by HP (we made hundreds of samples of different TZP-WC by HP and never found any carbon, what we found was W2C at high sintering temperatures which then forms by a different mechanism without carbon formation already described by Haberko et al.). As there are some indications that carbides can be decomposed during SPSwe are quite confident that the tentative explanation is correct. We also checked for some additional literature on the decomposition mechanisms in the TZP-WC system and found some supporting evidence.

We checked such flake like deposits by RAMAN and they turned out to be carbon. As a mechanical engineering institute our analytical capacities are limited, typically this could be addressed by TEM. In fact we must admit that further studies are required to prove this presumptive evidence.

The microstructure of the ceramic visible below the black spot has rounded shape while the fracture surface material in the surrounding unaffected material shows well crystallized. As the crack split the dark region and there is definitively an easy to cleave separate phase which separates the microstructure below and above the carbon deposit. Such rounded/flattened surfaces were also found in a recent paper in contact area between graphene and zirconia (17.     Obradović, N. Kern, F. Properties of 3Y-TZP zirconia ceramics with graphene addition obtained by spark plasma sintering, Ceram. Int. 2018, 44, 16931-16936.)